# Evaluation of the Effects of Human Dental Pulp Stem Cells on the Biological Phenotype of Hypertrophic Keloid Fibroblasts

**DOI:** 10.3390/cells10071803

**Published:** 2021-07-16

**Authors:** Ming Yan, Ling-Ling Fu, Ola A. Nada, Li-Ming Chen, Martin Gosau, Ralf Smeets, Hong-Chao Feng, Reinhard E. Friedrich

**Affiliations:** 1Department of Oral and Maxillofacial Surgery, University Medical Center Hamburg-Eppendorf, 20246 Hamburg, Germany; cnming.yan@hotmail.com (M.Y.); fu.lingling@hotmail.com (L.-L.F.); ola.abdulmonem@gmail.com (O.A.N.); m.gosau@uke.de (M.G.); r.smeets@uke.de (R.S.); r.friedrich@uke.de (R.E.F.); 2Department of Oral and Maxillofacial Surgery, Hebei Eye Hospital, Xingtai 054000, China; 3Department of Oral and Maxillofacial Surgery, Guiyang Hospital of Stomatology, Guiyang 050017, China; cnliming.chen@hotmail.com; 4Department of Oral and Maxillofacial Surgery, Division of “Regenerative Orofacial Medicine”, University Medical Center Hamburg-Eppendorf, 20246 Hamburg, Germany

**Keywords:** human dental pulp stem cells, fibroblast, co-culture, keloid

## Abstract

Objective: Despite numerous existing treatments for keloids, the responses in the clinic have been disappointing, due to either low efficacy or side effects. Numerous studies dealing with preclinical and clinical trials have been published about effective therapies for fibrotic diseases using mesenchymal stem cells; however, no research has yet been reported to scientifically investigate the effect of human dental pulp stem cells (HDPSCs) on the treatment of keloids. The objective is to provide an experimental basis for the application of stem cells in the treatment of keloids. Methods: Human normal fibroblasts (HNFs) and human keloid fibroblasts (HKFs) were cultured alone and in combination with HDPSCs using a transwell cell-contact-independent cell culture system. The effects of HDPSCs on HKFs were tested using a CCK-8 assay, live/dead staining assay, quantitative polymerase chain reaction, Western blot and immunofluorescence microscopy. Results: HDPSCs did not inhibit the proliferation nor the apoptosis of HKFs and HNFs. HDPSCs did, however, inhibit their migration. Furthermore, HDPSCs significantly decreased the expression of profibrotic genes (CTGF, TGF-β1 and TGF-β2) in HKFs and KNFs (*p* < 0.05), except for CTGF in HNFs. Moreover, HDPSCs suppressed the extracellular matrix (ECM) synthesis in HKFs, as indicated by the decreased expression of collagen I as well as the low levels of hydroxyproline in the cell culture supernatant (*p* < 0.05). Conclusions: The co-culture of HDPSCs inhibits the migration of HKFs and the expression of pro-fibrotic genes, while promoting the expression of anti-fibrotic genes. HDPSCs’ co-culture also inhibits the synthesis of the extracellular matrix by HKFs, whereas it does not affect the proliferation and apoptosis of HKFs. Therefore, it can be concluded that HDPSCs can themselves be used as a tool for restraining/hindering the initiation or progression of fibrotic tissue.

## 1. Introduction

During wound healing, a dynamic balance of synthesis and the degradation of collagen usually results in either a physiological or a pathological scar. A pathological scar can be described as a fibroproliferative disease caused by the hyperproliferation of fibroblasts and excessive synthesis of the extracellular matrix (ECM) in the process of dermis wound healing following skin trauma or serious burns [1,2,3]. Moreover, pathological scars can be classified into the following two types: hypertrophic scar and keloid. The former is predominantly localized above the original wound region, with a reddish or pinkish appearance, which can sometimes be pruritic. Furthermore, a hypertrophic scar usually regresses after years to form a matured scar [4,5], whereas the situation is strikingly different when the latter is involved. Although a keloid is a non-malignant disease, keloids almost always overgrow onto the surrounding skin, where they can often lead to malignant manifestations such as pain, ulceration, secondary infection, active angiogenesis and even carcinogenesis [6]. Such symptoms not only impact the quality of life, but they can also be highly unsightly, contributing further to psychological disturbances and corresponding social distress.

Despite numerous existing treatments for keloids, such as surgical removal, hormonal therapy, laser treatment or radiotherapy, as well as interventional sclerotherapy, the responses in the clinic have been disappointing due to either low efficacy or side effects, or a combination of both [7]. In order to address the aforementioned existing and urgent clinical need, new and more efficient therapeutic strategies are in order.

Numerous studies dealing with preclinical and clinical trials have been published about effective therapies for fibrotic diseases using mesenchymal stem cells, including the following: bone-marrow-derived mesenchymal stem cells (BMSCs) [8,9,10] and adipose derived mesenchymal stem cells (ADSCs) [11]. Human dental pulp stem cells (HDPSCs) are a type of adult stem cell that possess self-renewal, self-replication and multi-differentiation properties, where they can differentiate into a variety of mesodermal tissue cells, such as chondrocytes, osteoblasts, cardiomyocytes as well as adipocytes [12]. However, no research has yet been reported to scientifically investigate the effect of HDPSCs on the treatment of keloids.

The objective is to provide an experimental basis for the application of stem cells in the treatment of keloids. A co-culture method was set up to investigate the influence and mechanism of dental pulp stem cells on keloid fibroblast properties, such as cell proliferation, migration, collagen synthesis, invasion and apoptosis.

## 2. Materials and Methods

### 2.1. Cell Isolation and Cultures

Isolation of human dental pulp stem cells was performed using the explant growth method. Extracted teeth were obtained from patients (aged 18–25 years) with informed consent at the Guiyang Hospital of Stomatology (Guiyang, China). The retrieved dental pulp tissue was cut with sterilized scissors into small pieces and the tissue pieces were then plated onto 100-mm cell culture dishes in DMEM with 10% fetal bovine serum (FBS) and penicillin (100 U/mL)/streptomycin (100 μg/mL) at 37 °C and 5% CO_2_. Cells were further grown until confluence was reached. Human dental pulp stem cells were obtained using a limiting dilution method, a method previously established by our group [13]. Only passages 3–5 of dental pulp stem cells were used in this study.

Seven tissue samples (0.5 cm^3^) of keloids and seven corresponding normal skin tissue samples were used as a reference (Table 1). All tissue samples were taken from the maxillofacial area of patients in the same hospital with informed consent. The samples were classified into 2 groups (keloid and normal skin) based on clinical and pathological diagnosis. Furthermore, pathological characteristics were diagnosed upon staining with Hematoxylin and Eosin (H&E) with the following findings: (1) the dermal layer of the skin is markedly thickened; (2) disappearance of skin appendages; (3) collagen fibers begin to coarse into thick bundles. Finally, fibroblasts from keloid and normal skin isolation were also performed using the explant growth method. Samples were cut into 0.5 mm^3^ tissue pieces. Each tissue piece was then cultivated onto 100-mm cell culture dishes in DMEM with 10% fetal bovine serum (FBS) and penicillin (100 U/mL)/streptomycin (100 μg/mL) at 37 °C and 5% CO_2_. Cells were harvested until confluence was reached. Cells from passages 3–5 were used for this study.

### 2.2. Flow Cytometry Was Used to Detect the Surface Markers of HDPSCs

HDPSCs at passage 3 were trypsinized by 0.25% trypsin, washed twice with PBS and resuspended at a concentration of 1 × 10^9^/L in culture medium, according to the minimum criteria set by The International Society for Cellular Therapy in 2006 for defining multipotent MSCs [14]. The antibodies used in the experiments were CD19 (561295; BD Biosciences, San Jose, CA, USA), CD44 (560531; BD Biosciences, San Jose, CA, USA), CD45 (560368; BD Biosciences, San Jose, CA, USA), CD90 (555595; BD Pharmingen, San Jose, CA, USA), HLA-DR (560652; BD Biosciences, San Jose, CA, USA), CD29 (17-0299; eBiosciences, San Jose, CA, USA), CD73 (561014; BD Biosciences, San Jose, CA, USA), CD105 (560819; BD Biosciences, San Jose, CA, USA), CD73 (561014; BD Biosciences, San Jose, CA, USA), CD34 (560940; BD Biosciences), CD105 (560819; BD Biosciences) and CD11b (11-0113; eBiosciences, San Jose, CA, USA). In flow cytometry tubes, 1 mL of the cell suspension was collected with 5 μL of each of the anti-fluorescein isothiocyanate-conjugated antibodies. Cells were incubated at 4 °C for 1 h in the dark, and the cell surface markers were then analyzed on a BD Fortessa (BD Biosciences) instrument. All data analysis was conducted using FlowJo (v.10) software (FlowJo, Ashland, OR, USA).

### 2.3. Transwell Co-Culture Systems

In our project, 6-well Transwell systems, where cells shared medium without making any contact with transwell inserts (0.4-micrometer pore size), were employed (Figure 1A). Culture media was composed of DMEM with the following constituents: 10% FBS, 1% glutamine and 1% penicillin/streptomycin. There were the following four culture conditions in this project: (Figure 1B) HNFs monoculture, (Figure 1C) HDPSCs/HNFs co-culture, (Figure 1D) HKFs monoculture and (Figure 1E) HDPSCs/HKFs co-culture. In the monoculture groups, HNFs or HKFs were seeded in the bottom chambers. In the co-culture groups, HNFs or HKFs were seeded into the bottom chambers, and HDPSCs into the top chambers. The ratio of cells in the upper chamber to cells in the lower chamber was 1:1.

### 2.4. Cell Morphology

Cells were photographed under an inverted microscope (ECLIPSE Ts2-FL, Nikon, Japan) at days 1, 3, 5, and 7.

### 2.5. Cell Proliferation

Both normal and keloid fibroblasts were each individually seeded into 6-well plates at a concentration of 1 × 10^4^ cells per well. In this study, cellular proliferation was analyzed using the CCK8 assay (Dojindo, Kumamoto, Japan) according to manufacturer’s instructions. After removal of the supernatant, 660 μL of CCK8 solution (60 μL CCK8:600 μL medium) was added to each well and incubated in the incubator for 2 h on days 1, 3, 5 and 7. Then, a 100-microliter volume of the supernatant was separated by centrifugation and transferred into a fresh 96-well plate. Readings at wavelengths of 450 nm were recorded using a Fluostar Omega plate reader, and a standard curve was then plotted against the readings of the standards.

### 2.6. Collagen Synthesis Detection

Both normal and keloid fibroblasts were each individually seeded into 6-well plates at a concentration of 5 × 10^5^ cells per well. Supernatants were retrieved from 6-well plates after 48 h when cells reached confluence and were handled according to the instructions of Hydroxyproline Assay Kit (MAK008; Sigma Aldrich, St. Louis, MI, USA). A total of 1.5 mL of the supernatant was harvested and resuspended in 0.05 mL of digestion solution at 37 °C for 3 h. Then, 0.5 mL of solution A was transferred into the mixture for 10 min and 0.5 mL of solution B was added and allowed to react for 5 min, followed by the addition of 1 mL of solution C at 60 °C for 15 min. Absorbance was then read at a wavelength of 550 nm. Hydroxyproline concentrations were calculated from a standard curve of hydroxyproline (0–100 mg/mL).

### 2.7. Wound Scratch Assay to Detect the Cell Migration Ability

Normal and keloid fibroblasts were seeded individually into 6-well plates at a concentration of 5 × 10^5^ cells per well. When the cells reached 90% confluency, 200 µL sterile pipette tips were used to scratch the bottoms of the 6-well plate culture wells. After rinsing out the floating cells using PBS, cells were starved for 24 h in serum-free medium. In the test group, transwell chambers were seeded with a concentration of 5 × 10^5^ of dental pulp stem cells. The wound area was recorded using a light microscope camera after scratching at 0 and 24 h. Finally, the rate of cellular migration was quantified using image J software V1.51 (NIH software, Bethesda, MD, USA). The migration rate of cells was measured using the following formula: (W_0h_ − W_24h_)/W_0h_ × 100%.

### 2.8. Live and Dead Staining

Live and dead cells were assessed using LIVE/DEAD^®^ staining kits (Cat# L3224; Molecular Probes, Eugene, OR, USA) according to the manufacturer’s instructions, where 500 µL of staining solution was added to each well after the chamber was removed at days 1, 3, 5 and 7, the cells were then incubated in the dark at room temperature for 10 min, and then photographed with the employment of immunofluorescence microscopy. Five images, one from the center and four from the periphery, were obtained from each well using an Olympus inverted fluorescence microscope. Live and dead cells were counted using Image J software, and the rates of cell survival were calculated as Live cells/(Live + Dead cells)%.

### 2.9. Western Blot

Normal and keloid fibroblasts were seeded in six-well plates at a density of 5 × 10^5^ cells/well and transwell chambers were seeded with a concentration of 5 × 10^5^ of dental pulp stem cells cultured in complete medium for 24 h. The medium was discarded and the fibroblast in the lower chamber washed 3 times with phosphate buffered saline (PBS). Total cell lysates from different groups were obtained by lysing the cells in a RIPA buffer (Beyotime Institute of Biotechnology, Shanghai, China) containing protease inhibitors (100 µM phenylmethylsulfonyl fluoride, 10 µM leupeptin, 10 µM pepstatin and 2 mM EDTA). The protein content was quantitated using a BCA Protein Assay kit (Beyotime Institute of Biotechnology). Proteins (50 µg/lane) were separated via 4–20% SDS-PAGE and transferred to nitrocellulose membranes (Pall Life Sciences, Pensacola, FL, USA). Membranes were blocked with 2% Bovine Serum Albumin (BSA) for 1 h at room temperature and probed overnight at 4 °C with primary antibodies against CTGF (ab6992), α-SMA (ab7817), TGF-β1 (ab64715), TGF-β2 (ab66045) and collagen I (ab292), which were products of Abcam (Cambridge, UK), in a humidified chamber. Membranes were washed and incubated with horseradish peroxidase-conjugated secondary antibodies (1:5000) (cat. nos. AS014 and AS003; ABclonal Biotech Co., Ltd., Woburn, MA, USA) at room temperature for 2 h, and were visualized using an enhanced chemiluminescence system (EMD Millipore, Billerica, MA, USA). Fluorescent secondary antibodies were then added and immunoblots were thereafter imaged with a two-channel (at 700 plus 800 nm) IR fluorescent Odyssey CLx imaging system (LI-COR^®^, Lincoln, NE, USA). Results were quantified using image J software.

### 2.10. Real-Time Quantitative PCR (RT-qPCR)

Normal and keloid fibroblasts were seeded in six-well plates at a density of 5 × 10^5^ cells/well and transwell chambers were seeded with dental pulp stem cells at a concentration of 5 × 10^5^ and cultured in complete medium for 24 h. The medium was discarded and the fibroblast in the lower chamber was washed 3 times with phosphate buffered saline (PBS). Thereafter, cells were harvested using a cell scraper after the addition of a lysis buffer. RNA was consequently extracted through the following steps: lysing cells in Trizol reagent (Cat No. 15596-026, Life Technologies, Carlsbad, CA, USA), followed by extracting RNA in trichloromethane, and then precipitating it in isopropanol, and finally resuspending it in RNase-free water. The RNA concentration and purity levels were determined using a Nanodrop2000 Spectrophotometer (Thermo Fischer Scientific, Waltham, MA, USA). Total RNA (2.5 μg) was subjected to cDNA synthesis using a qScript cDNA SuperMix (Quanta BioSciences, Beverly, MA, USA) through the following consequent cycles: firstly at 25 °C for 5 min, followed by 42 °C for 30 min and finally at 85 °C for 5 min. A real-time PCR was performed to determine the mRNA levels of TGF-β1, fibrinogen, α-SMA and GAPDH using SYBR Green Master MIX (ABI, Vernon, CA, USA). For a relative mRNA expression, the 2∆∆cq method, in which ΔCq = each corresponding Cq value − minimum Cq value, was calculated.

### 2.11. Statistical Analyses

The data are shown as the mean ± standard deviation (SD) from at least three independent repeated experiments. Student *t*-test was used to analyze the differences in mean values between the two groups. Significant differences were defined as *p* < 0.05. All statistical analyses were performed using Graph Pad Prism software (Graph Pad Prism, San Diego, CA, USA; RRID:SCR_002798) version 7.0a.

## 3. Results

### 3.1. Clinical and Pathological Characteristics

The samples were classified into two groups (keloid and normal skin). Compared with normal skin tissue, the keloid is harder in texture, possessing either a round or oval structure, and involving excessive growth beyond the boundary of the originally wounded skin, thereby invading neighboring normal tissues. See Figure 2.

### 3.2. Identification of Stem Cells

To confirm that these HDPSCs were of mesenchymal origin, cell surface markers were detected using flow cytometry to determine the mesenchymal origin of the HDPSCs. The following markers were positively expressed at the respective percentages: CD29 at 99.30%, CD73 at 99.60%, CD105 at 99.50% and CD90 at 99.60%, whereas the hemopoietic stem cell marker, CD19 at 0.43%, CD44 at 0.17%, CD45 at 0.74%, HLA-DR at 0.14%, CD34 at 0.11% and CD11b at 0.19%. See Figure 3.

### 3.3. HDPSCs Did Not Inhibit the Proliferation of HNFs and HKFs

Compared to the HKFs cultured alone, the CCK-8 assay showed that no difference in the proliferation of HKFs was observed in the HDPSCs/HKFs co-culture group at days 1, 3, 5, and 7 (*p* > 0.05, *t*-test; *n* = 7, mean ± SD). Moreover, as a positive control, there was also no detected difference in the proliferation of HNFs in the DPSCs/HNFs co-culture group at days 1, 3, 5, and 7 (*p* > 0.05, *t*-test; *n* = 7, mean ± SD). However, the proliferation rate of HKFs was much higher than that of HNFs on the third day (Figure 4A). Compared with the control group, the morphology of the fibroblast was still elongated, and there was no variation between the two groups.

### 3.4. HDPSCs Did Not Influence the Apoptosis of HNFs and HKFs

Our results indicate that the cellular apoptosis of HNFs and HKFs was not affected by HDPSCs. As shown in Figure 5, with respect to cell apoptosis, both HKFs and HNFs were not affected through their coculture with HDPSCs (*p* > 0.05). This result further validates the cell proliferation results.

### 3.5. The Effects of HDPSCs on the Migration of HKFs and HNFs

The wound scratch assays showed that there was only a difference in the cell migration ability in the co-culture group of the human keloid fibroblast after 24 h of culture (*p* < 0.05). As a positive control, cellular migration of the HNFs remained uninfluenced (*p* > 0.05). See Figure 6.

### 3.6. Inhibited Expression of Fibrosis-Associated Gene Phenotype and Protein Expression in HKFs and KNFs

Compared with solely cultured cells, only the expression levels of CTGF in HNFs were observed to be unaffected (*p* > 0.05, *t*-test; *n* = 7, mean ± SD), whereas the expression levels of CTGF, TGF-β1 and TGF-β2 in HKFs and KNFs at the mRNA and protein levels were significantly inhibited when co-cultured with the HDPSCs (*p* < 0.05, *t*-test; *n* = 7, mean ± SD). These findings indicate that both the transcriptional and posttranslational levels were inhibited. See Figure 7.

### 3.7. HDPSCs Inhibits Extracellular Matrix Synthesis of HKFs and HNFs

Compared with solely cultured cells, the expression levels of collagen I in HNFs were observed to be unaffected (*p* > 0.05, *t*-test; *n* = 7, mean ± SD), whereas the expression levels of collagen I and α-SMA in HKFs and KNFs at the mRNA and protein levels were significantly inhibited when co-cultured with the HDPSCs (*p* < 0.05, *t*-test; *n* = 7, mean ± SD). These findings indicate that both the transcriptional and posttranslational levels were inhibited. See in Figure 8A–D,F.

There was a significant decrease in the resultant hydroxyproline concentrations of the HKFs co-culture group (*p* < 0.05, *t*-test; *n* = 7, mean ± SD), whereas no significant changes were detected in the other three groups (*p* > 0.05, *t*-test; *n* = 7, mean ± SD). See in Figure 8E.

## 4. Discussion

Wound repair is a complex process that often leads to the formation of scars following traumatic skin injuries. This process is associated with the functions of various cells, such as fibroblasts, endothelial cells, macrophages and lymphocytes [15,16,17], among which the biological behavior of fibroblasts is considered to be a key factor in the scar formation process. Numerous studies have shown that fibroblasts, which happen to be the main constituents of keloid tissue, have the ability to over proliferate and are accompanied by incomplete apoptosis, together with the abnormal synthesis of collagen, which overall results in the continuous proliferation of keloid tissue [18,19]. Therefore, the inhibition of fibroblast proliferation and induction of apoptosis in keloid tissues can majorly reduce keloid tissue proliferation and thereby delay disease progression, which is important for the improvement and treatment of keloid scars. Therefore, understanding the biology of keloid fibroblasts is important for the treatment of keloids.

Mesenchymal stem cells (MSCs) are an important member of the stem cell family, which have been used to treat scar formation-related diseases such as pulmonary fibrosis. Moreover, great progress has been made where they have also been used to inhibit cardiac scar formation through the secretion of various cytokines [20]. Studies have shown that stem cells have been used in animal models and in a few clinical trials for the regeneration of diseased organs. Furthermore, stem cells have been shown to improve tissue repair by secreting interleukins such as interleukin 6, interleukin 8, interleukin 10 and other proteins that are suitable for inducing tissue regeneration [21,22]. Recent studies have shown that adipose-derived stem cells are able to inhibit mRNA expression levels of COL1A1, transforming growth factor β1, connective tissue growth factor and alpha actin 2 (ACTA2) in renal fibrosis tissues, thereby playing a therapeutic role in renal fibrosis [23]. Whether or not dental pulp stem cells can play a role in the repair of skin scars by inhibiting the proliferation and migration of keloid fibroblasts through the secretion of cytokines has not been clearly reported thus far.

Dental pulp stem cells, human keloid fibroblasts and human normal skin fibroblasts were isolated using the tissue explant method. The molecular mechanisms of action of dental pulp stem cells on the proliferation, migration and apoptosis of keloid fibroblasts were investigated using a transwell co-culture. The histology of keloid scars is highlighted by the secretion and deposition of large amounts of extracellular collagen [24]. Furthermore, both skin keloids and fibrous tumors are pathologically fibrous connective tissue lesions with large extracellular collagen deposition [25]. In this study, we also found that the co-culture of HDPSCs inhibited the migration of HKFs and HNFs but did not inhibit the proliferation of either HKFs and HNFs, nor did it induce the apoptosis of HKFs and HNFs. Our experimental data are similar to the results of a recent study reporting that the conditioned medium and cell lysates of human-derived WJ-MSCs inhibit the migration of human HKFs [26]. In comparison with this report, we used HDPSCs, which have the advantages of being easily accessible with minimal patient harm, and the fact that they can be used autologously. However, studies reporting opposing results also exist [27], in which the conditioned medium of human WJ-MSCs could promote HKFs proliferation in a paracrine manner through an indirect transwell co-culture treatment system. We hypothesize that the reasons for these different results are attributed to the different types of MSCs used, the different cell treatment cultures and different assays, all of which impact the following biological behaviors of keloid fibroblasts: migration, proliferation and collagen secretion, in addition to the apoptosis involved in wound repair after skin injury. Therefore, by inhibiting the migration and proliferation of keloid fibroblasts, keloid formation is also consequently inhibited [28]. Moreover, it has been shown that the inhibition of matrix metalloproteinase expression in keloid fibroblasts can inhibit the migration ability of keloid fibroblasts and, therefore, function as a keloid treatment aid [29]. Furthermore, similar findings have been reported where inhibition of mTOR protein expression in keloid fibroblasts can also inhibit the migratory ability of keloid fibroblasts and thereby further inhibiting the development of keloids [30]. Similarly, and consistent with the aforementioned studies, the results of the present study revealed that dental pulp stem cells are also able to inhibit the migration of keloid fibroblasts in vitro, thereby suggesting that dental pulp-derived stem cells can promote wound healing by inhibiting scar formation.

The expression of anti-fibrotic and pro-fibrotic genes is closely related to the pathogenesis of fibrotic diseases, which in another way confirms that hyperplastic scars and keloids are classified as fibrotic diseases. TGF-β1 and TGF-β2 overexpression is an important cause for excessive scar proliferation and fibrosis, and studies have shown that the targeted reduction in TGF-β1 and TGF-β2 expression in hyperplastic scars and keloids can inhibit scar proliferation and achieve clinical therapeutic effects [29,30]. Therefore, TGF-β1 and TGF-β2 have also become the targets of numerous studies that tackle the treatment of hyperplastic scars and keloids. Moreover, CTGF is a marker protein of fibrotic diseases in which it promotes both the proliferation of fibroblasts as well as the secretion and deposition of extracellular matrix proteins, such as collagen I and fibronectin [31,32]. In the current study, it was found that after 24 h of HDPSC co-culture with HKFs, both the gene and protein expression of TGF-β1, TGF-β2 and CTGF were significantly reduced in HKFs. This is in accordance with studies that have shown MSCs as being capable of secreting cytokines in order to alter some biological phenotypes of fibroblasts, such as fibrotic and proliferative phenotypes, through paracrine functions [33,34]. Increasing evidence thereby suggests that the paracrine function of MSCs is an important potential mechanism for their cellular therapeutic function. Thereupon, after being injected into the body, MSCs can, on the one hand, inhabit the area of tissue damage through the processes of chemotaxis, proliferation and, eventually, differentiation by evolving into the cell type required for the secretion of extracellular matrix that would be needed to repair the damage in the recipient area; on the other hand, MSCs, upon entering the body, can exhibit a paracrine function, which entails the secretion of cytokines and nutrient-active substances required for repairing the damage, and inducing the body’s self-generated cells to repair the tissue damage. The SMAD signaling pathway is a downstream mediator of TGF-β. After phosphorylation, the phosphorylation of R-SMAD 3 is upregulated in the keloid, whereas the downregulation of R-SMAD 3 significantly reduces procollagen gene expression in keloid fibroblasts. I-SMAD 6 and I-SMAD 7 inhibit the action of TGF-β. SMAD 6 also inhibits the binding of SMAD 4 and R-SMAD. The expression of I-SMAD 6 and 7 is reduced in keloid fibroblasts. By inhibiting the TGF-β1-SMAD signaling pathway and activating TLR7 or SMAD 7, keloid formation can be significantly reduced. The toll-like receptor signaling pathway plays a protective as well as a destructive role. After skin injury, toll-like receptors (TLR) are combined as damage-related molecular patterns (DAMP) to enable the innate immune system to respond to sterile tissue damage. As the concentrations of several pro-inflammatory and pro-fibrotic cytokines in macrophages increase in response to TLR stimulation in macrophages, fibroblast gene expression and TGF-β responses change, leading to an increased collagen production.

A very distinctive feature of proliferative scars and keloid tissues is the excessive deposition of the extracellular matrix [35]. There are two main mechanisms that can lead to an excessive extracellular matrix deposition, one being an increase in the extracellular matrix synthesis, and the other being a decrease in the extracellular matrix degradation. Compared to normal skin, the amount of collagen synthesis in hyperplastic keloid scars is three times higher, while keloid scars can reach up to 20 times higher [4]. In our experiments, it was further observed that due to the co-culture with DPSCs, both HSFs and HKFs displayed a reduction in extracellular matrix synthesis, as shown by the reduced expression of collagen type I, α-SMA and hydroxyproline. Moreover, in the extracellular matrix of scar tissue, collagen type I is normally the predominantly present collagen type. Furthermore, the detection of hydroxyproline content in cell culture media is recognized as a reliable indicator of the ability of fibroblasts to synthesize collagen. Relevantly, it was previously reported that both bone marrow MSCs and dermal MSCs were able to inhibit collagen synthesis and the expression of α-SMA in keloid fibroblasts [27].

Finally, our findings are consistent with previous studies by showing that dental pulp stem cells were able to inhibit the expression of pro-fibrotic genes and keloid fibroblast collagen synthesis in vitro. Furthermore, this study revealed that dental pulp stem cells inhibit the migration of keloid fibroblasts. Nevertheless, the specifically secreted cytokines that influence the biological behavior of keloid fibroblasts need to be further investigated by subsequent experiments.

## 5. Conclusions

The co-culture of HDPSCs inhibits the migration of HKFs and the expression of pro-fibrotic genes, while promoting the expression of anti-fibrotic genes. HDPSCs’ co-culture also inhibits the synthesis of the extracellular matrix by HKFs, whereas it does not affect the proliferation and apoptosis of HKFs. Therefore, it can be concluded that HDPSCs can be used as a tool for restraining/hindering the initiation or progression of fibrotic tissue.

## Figures and Tables

**Figure 1 cells-10-01803-f001:**
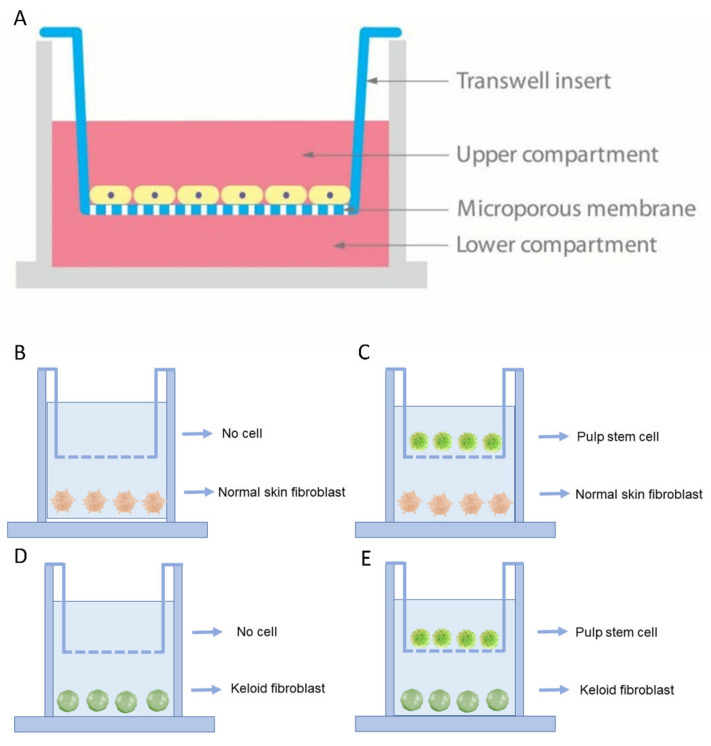
Study design. (**A**): A schematic design of the Transwell co-culture model was established using Transwell chambers with a 0.4-micrometer pore size that allows for the passage of chemical and biochemical molecules. (**B**–**E**): 4 combinations of the inducing and differentiating cells.

**Figure 2 cells-10-01803-f002:**
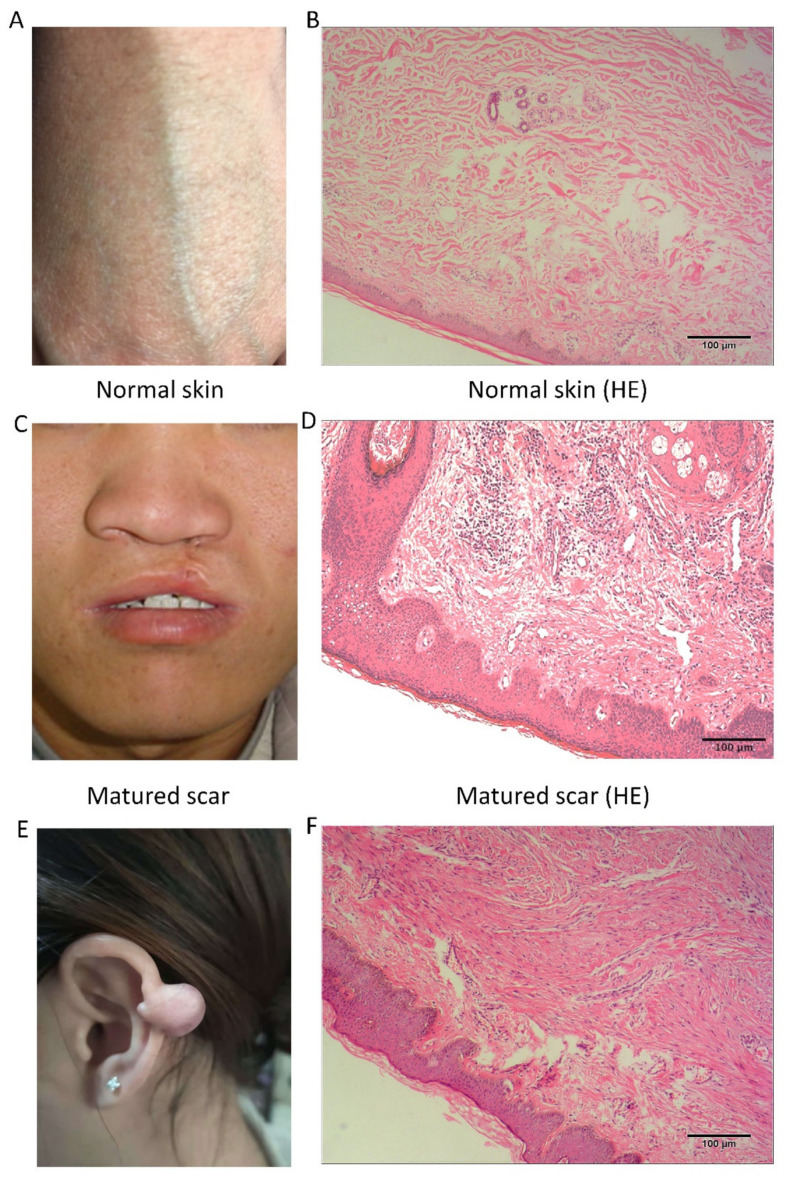
Gross and hematoxylin and eosin (H&E) staining image of normal skin, matured scar and keloid. (**A**) Normal skin. (**B**) H&E staining of normal skin. (**C**) Matured scar. (**D**) H&E staining of matured scar. (**E**) Keloid. (**F**) H&E staining of keloid.

**Figure 3 cells-10-01803-f003:**
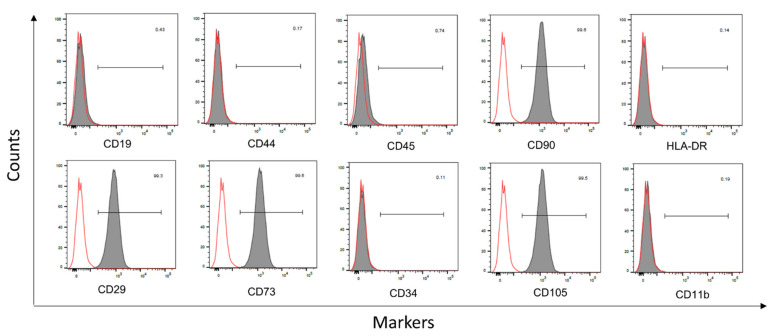
Flow cytometric results of passage 3 HDPSCs. Representative histograms showing antigen expression in bone-marrow-derived MSC. From left to right CD19, CD44, CD45, CD90, HLA-DR, CD29, CD73 CD105, CD73, CD34, CD105 and CD11b. Black-filled histogram: antigen expression; solid red line: auto-fluorescence control.

**Figure 4 cells-10-01803-f004:**
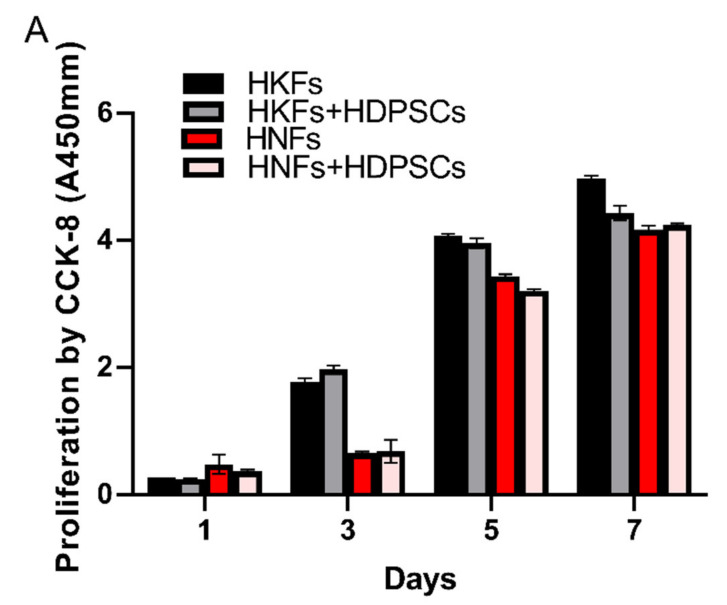
(**A**): The cell counts (proliferation) were analyzed using CCK-8 kits (Dojindo) on days 1, 3, 5, and 7. (**B**): The morphology of the fibroblasts was determined on days 1, 3, 5, and 7 by direct observation with a light microscope. Scale bar: 200 μm. HNFs: human normal fibroblasts, HKFs: human keloid fibroblasts, HDPSCs: dental pulp stem cells.

**Figure 5 cells-10-01803-f005:**
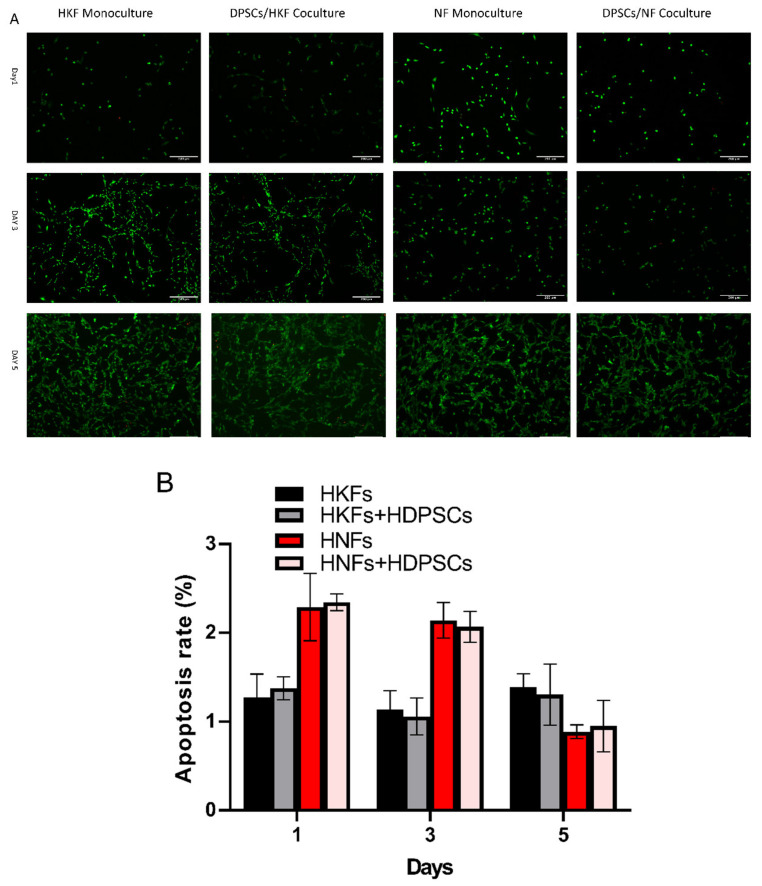
Apoptosis rate of HNFs and HKFs in mono and co-cultures. HDPSCs do not induce apoptosis in HNFs or HKFs. (**A**): Vital cells stain green with Calcein-AM, while dead cells stain red with propidium iodide on days 1, 3, and 5. Scale bar: 200 μm. (**B**): The quantification of cell apoptosis/necrosis using the percentage of PI-positive cells/AM-positive cells. (Mean q-SD, *n* = 5, Student *t*-test, *p* < 0.05).

**Figure 6 cells-10-01803-f006:**
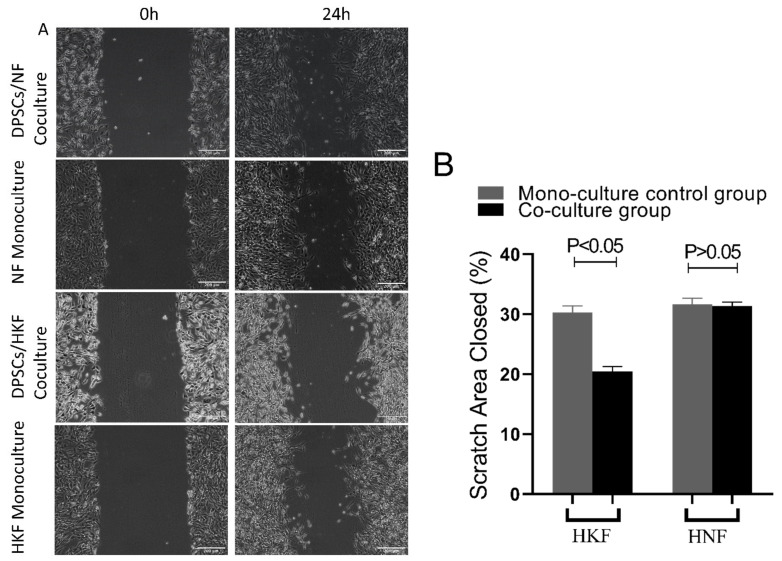
Cellular migration in scratch area. (**A**): Pictures were taken at 0 and 24 h, a magnification of 100× was used. (**B**): Percentage of wound area recovery by migrated cells was quantified by Image J. Significant differences were only detected in the co-culture group of the human keloid fibroblast after 24 h (*p* < 0.05, *t*-test; *n* = 7, mean ± SD).

**Figure 7 cells-10-01803-f007:**
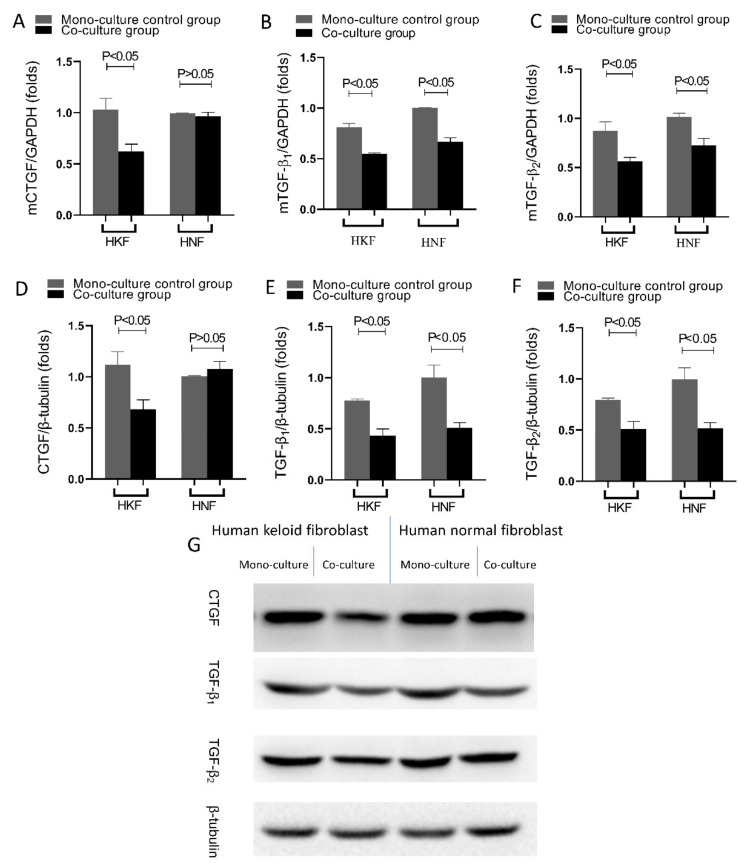
HDPSCs attenuated the pro-fibrotic phenotype of HKFs and HNFs. After 24 h of culture, cells were subjected to RT-qPCR and Western blot. Quantification of (**A**) CTGF, (**B**) TGF-β1 and (**C**) TGF-β2 gene expression, normalized to GAPDH expression (**D**–**F**). Quantification of (**D**) CTGF, (**E**) TGF-β1 and (**F**) TGF-β2 protein levels, normalized to β-tubulin expression. (**G**) Gels were analyzed by immunoblotting using the indicated antiserum.

**Figure 8 cells-10-01803-f008:**
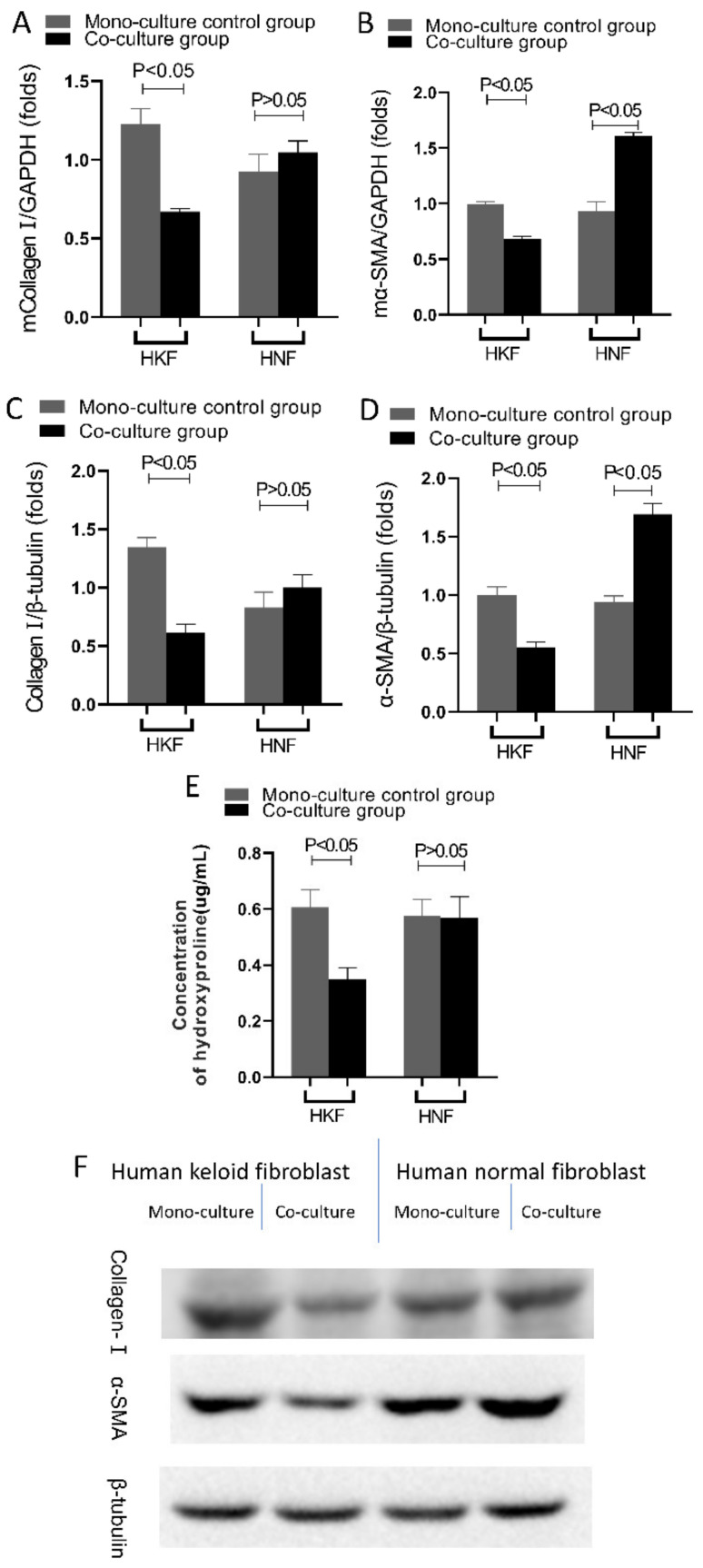
HDPSCs inhibits extracellular matrix synthesis of HKFs and HNFs. (**A**) After 24 h of culture and then subjected to RT-qPCR and Western blot. Quantification of (**A**) Collagen I, and (**B**) α-SMA gene expression, normalized to GAPDH expression. Quantification of (**C**) Collagen I, and (**D**) α-SMA protein levels, normalized to β-tubulin expression levels (**E**) Cell culture supernatants were collected and tested for hydroxyproline content. (**F**) Gels were analyzed by immunoblotting using the indicated antiserum.

**Table 1 cells-10-01803-t001:** Patient epidemiological data.

Subject	Age	Sex
1	29	F
2	38	M
3	21	M
4	32	M
5	27	F
6	25	F
7	33	M

## Data Availability

The data are available from the corresponding author on reasonable request.

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
