# Peer review of "Evaluation of the Effects of Human Dental Pulp Stem Cells on the Biological Phenotype of Hypertrophic Keloid Fibroblasts"

_cells, 2021, doi:10.3390/cells10071803_

Round 1

Reviewer 1 Report

Manuscript by Ming Yan et al. presents interesting and useful results of an experiment with a very interesting aspect concerning on dental pulp stem cells and keloid fibroblasts co-culture. However, in my opinion, there are some significant shortcomings in the work.

  1. The methodology part does not describe what statistical tests were used by the authors and whether these tests were adequate to the number of performed biological repetitions?
  2. Do the authors believe that 3 replicates in samples taken from patients (regardless of whether they were subsequently in vitro cultured or not) are sufficient for the obtained results to be statistically significant?
  3. How the group of donors / patients was selected - only the age range was given, so what was the sex, other diseases, medications taken?
  4. The entire methodology is devoid of references - does it mean that these are all author's (validated) methods?
  5. The description of the figure 1 is duplicated, please rewrite it.
  6. Figure 4B – it should be completed with the value of the cell density in a field unit.
  7. Figure 7F – WB for CTGF, the first band is much stronger than the others, what is not visible in the figure 7C.
  8. The literature used in the introduction and discussion points to the very poor preparation of the subject review. Used articles are few and old, although there are really many publications available even from 2020
  9. The conclusion in abstract and in the end of the text is a summary of the results, it does not provide conclusions nor the potential for using this data.
  10. Also some of the sentences are clunky (e.g. lines: 46-48; 60-61), it will good if a proficient native speaker can look at the text.

Author Response

  1. The methodology part does not describe what statistical tests were used by the authors and whether these tests were adequate to the number of performed biological repetitions?

Reply: we rewrite this part, please read this part in new manuscript.

  1. Do the authors believe that 3 replicates in samples taken from patients (regardless of whether they were subsequently in vitro cultured or not) are sufficient for the obtained results to be statistically significant?

Reply: Yes, we believe it have statistically meaning. The data are shown as the mean ± standard deviation (SD) from at least three independent repeated experiments. AND seven tissue samples (0.5cm3) of keloids and seven corresponding normal skin tissue samples were used as a reference.

  1. How the group of donors / patients was selected - only the age range was given, so what was the sex, other diseases, medications taken?

Reply: Exclusion criteria for scar specimens

Tissue samples of patients with the following conditions cannot be included:①The disease course is less than 6 months;②There is infection at the lesion site;③Have had radiotherapy, steroid injection therapy, etc.;④The patient also has other genetic diseases;⑤The specimen does not meet the clinical and pathological diagnostic criteria.

Before tissue specimen collection

Obtain the patient’s informed consent, and take photos and record the patient’s gender, age, disease, scar history, location of the sample and other relevant information.

The subcutaneous fat is removed from the specimen, and the tissue of about 1.0 cm×1.0 cm×0.5 cm in the middle part of the skin or scar is cut.

All tissues was used for the separation and culture of fibroblasts immediately after being taken,Or quickly put it in a freezing tube and store it in a low-temperature refrigerator at -80 ℃.

  1. The entire methodology is devoid of references - does it mean that these are all author's (validated) methods?

Reply: I will add the corresponding literature references

  1. The description of the figure 1 is duplicated, please rewrite it.

Reply: Rewrite it as your suggestion.

  1. Figure 4B – it should be completed with the value of the cell density in a field unit.

Reply: we submit the new figure

  1. Figure 7F – WB for CTGF, the first band is much stronger than the others, what is not visible in the figure 7C.

Reply: First we need to apologize. After all the authors' verification, there were two errors in this picture. The first error is a label error, there are two C label.  The second error is the western picture intercept error. We have corrected the picture, figure D is the corresponding the first row of figure F about western blot.

Once again, we apologize for the carelessness in our drawing process.

  1. The literature used in the introduction and discussion points to the very poor preparation of the subject review. Used articles are few and old, although there are really many publications available even from 2020

Reply: we change some new references

  1. The conclusion in abstract and in the end of the text is a summary of the results, it does not provide conclusions nor the potential for using this data.

Reply: we rewrite this part

  1. Also some of the sentences are clunky (e.g. lines: 46-48; 60-61), it will good if a proficient native speaker can look at the text.

Reply: Our colleague Susan has re-read the manuscript and polished it up. She was born in England and grew up in England

Reviewer 2 Report

Overall, the paper is on an interesting topic, but it needs a bit of work in all areas.

Specific comments

  1. Can you show that you stem cells are secreting some of the cytokines that promote repair? You have shown results from the fibroblasts but not the stem cells.
  2. Can you show how the stem cells are altering the fibroblasts. For example, are they acting through the SMAD signalling pathway as has been shown for ADSCs?
  3. Can you provide more information on the samples. Collate this information in a table with the age, sex, how long they had the keloid scar for (where relevant) and the exact region the sample was taken from. This can then go in the methods in a section called patient samples. Also all of your figure legend need to have the exact number of samples used in each experiment and the statistical test that has been applied.
  4. In your introduction or discussion can you provide an idea as to how this might work as a therapeutic on keloids that have already formed? (I think this is what you are suggesting?). In the objective you mention it as a treatment for keloid scars and I am not sure how this will work, is it to prevent them forming?
  5. Overall the paper needs some work (but is fixable) on the text in all sections of the publication. I suggest that you use your reference 9 for a guide as to how to write the results section as its is currently very brief an lacking rationales and what has been done. I have put some comments below to help you edit the paper.
  6. In the abstract there are lots of abbreviations that have not been spelled out (you always do this the first place they are written).
  7. In the objective section of the abstract the last sentence is not clear. Do you want to provide an experimental basis or provide a novel proposal – it is one or the other. This is said again at line 71 in the introduction.
  8. Still in the abstract the methods section there is an ‘and’ missing from the list of methods and also immunofluorescence microscopy is missing.
  9. Introductions do not have figures in them. If you want to show some examples of the samples you are using then this should be in the results section.
  10. Line 61 Chemotherapeutics generally refer to drugs used to treat cancer. I think you probably want to say therapeutics.
  11. Line 64 is missing an ‘and’
  12. Line 62 is a little misleading as it says clinical trials have been published about effective therapies for fibrotic diseases using mesenchymal stem cells. You were talking about keloid scars where they have not been used clinically (see Bojanic 2021, Cell and Tissue Research) so be clear about what you mean. You also cite 2 studies that are done in vitro and not even in vivo in preclinical mouse models.
  13. It’s unclear from your methods exactly how the RT-PCR and western blotting were performed. Were the cells co-cultured with the stem cells in the upper chamber and fibroblast in the well, then the upper chamber removed and the fibroblast in the lower chamber analysed? There are lots of this sort of information missing so go carefully through it to make sure it is written clearly enough that someone could repeat the study.
  14. Line 82 did you also have antibiotics and a fungicide in your media?
  15. Line 92 ‘collagen fibers begin to coarse into thick bundles’ coarse means thick so maybe you are trying to say arrange as?
  16. Line 93 ‘Finally, fibroblasts from keloid and normal skin isolation were also performed using the explant growth methodology at passages 3-5.’ You need to describe this method better and what were they cultured in etc.
  17. Line 97 what were the cells resuspended in?
  18. Line 102 Analysed by flow cytometery. This needs more details, e.g. machine and what software you used.
  19. Line 105 you need to mention that it’s the Transwells that have the pores. The way this is written it reads as if the plates have pores. Also, the figure needs removing (Transwells are standard method) and the information should be described better in the methods text. For example you need to add how many cells were plated in the well and in the Transwell upper chamber for each cell type.
  20. Line 121. How did you analyse and define cell morphology?
  21. Line 124. You have not mentioned the cells and how many were plated etc. These details need to be added.
  22. Line 134, I think you need to say according to the instructions in this sentence.
  23. Line 136 0.5 ml ‘of’ solution A. Similarly, for the two lines below an ‘of’ needs adding.
  24. Line 143 ‘confluency ‘a’ 200ul sterile’
  25. 146 Transwell is a brand and its not written trans well
  26. Line 148, please explain better how you anlysed this. For example, did you use an imageJ plug in, if not what tool did you use? Are you measuring distance or area?
  27. Line 156 This method is not for a western blot, it looks like it is half immunofluorescence microscopy and half western blotting. Western blotting does not start with fixing cells in paraformaldehyde and permeabilising the cells like this but immunofluorescence microscopy has this in it. You have a section for western blotting and one for immunofluorescence microscopy.
  28. Line 169 you need to add details of the lysis buffer.
  29. Line 187 and all the sections below need some sort of lead into what you are doing and how you are doing this. Ref 9 is probably a good example for you to look at to see what I mean. At the moment we don’t know what you have done and why or what cells were used. This information must be added before you launch into the result.
  30. Figure 2 p6 you will need to provide higher resolution images these are fuzzy, and I cannot read the axis. Your figure legend also needs more detail about the experiment performed, number of samples analysed etc. Have a look at other papers and see how they present similar data.
  31. Line 199 again you need to say what you have done and maybe why you have co-cultured the cells so the person reading this knows what you are trying to do with this experiment. Same throughout the results.
  32. Figure 4 the key to the graph needs better labelling – co-culturing of what? You have only one cell type in the key. Please take a look at reference 9 and see how they have put together their figures. Again all your figure legends need work.
  33. Figure 4, the morphology part of the figure, I am not sure what I am supposed to be looking at and at the magnification provide it is hard to determine morphology. There is also no main text associated with this result so you have not told me why you have done this or what you found in your text section.
  34. The LIVE/DEAD® staining kit (Cat# L3224; Molecular Probes) you used does not measure apoptosis, only live or dead cells. You need to fix this in the text and in your figure as its not apoptosis being measured. I also cannot see any dead cells in those figures as the magnification is not high enough?
  35. Please provide quantification of all of the western blots.
  36. The discussion mentions ‘the results of the present study revealed that dental pulp stem cells are also able to inhibit the migration of keloid fibroblasts in vitro, thereby suggesting that dental pulp-derived stem cells can promote wound healing by inhibiting scar formation.’ Please add some more discussion around why this will inhibit scar formation.
  37. There are a few typographical errors that need fixing and some small issues with grammar.
  38. The discussion mentions ‘this study revealed the molecular mechanism by which dental pulp stem cells inhibit the proliferation and migration of keloid fibroblasts, and whether or not dental pulp stem cells play a regulatory role by secreting cytokines’. This is not what you have shown. Both the PCR and western blot are on the fibroblasts, not the stem cells, so you have not shown they secrete cytokines. You need to look carefully at your results and your interpretation of them.

I suggest all authors look at the publication before submitting as there are so many things missing that it detracts from the data.

Author Response

  1. Can you show that you stem cells are secreting some of the cytokines that promote repair? You have shown results from the fibroblasts but not the stem cells.

Reply:we did not test cytokines in this project, so we can not show. 

  1. Can you show how the stem cells are altering the fibroblasts. For example, are they acting through the SMAD signalling pathway as has been shown for ADSCs?

Reply:we only did TGFβ,If we pass the first round of review, we can do the SMAD signalling pathway

  1. Can you provide more information on the samples. Collate this information in a table with the age, sex, how long they had the keloid scar for (where relevant) and the exact region the sample was taken from. This can then go in the methods in a section called patient samples. Also all of your figure legend need to have the exact number of samples used in each experiment and the statistical test that has been applied.

all the keloid sample from earlobe

  1. In your introduction or discussion can you provide an idea as to how this might work as a therapeutic on keloids that have already formed? (I think this is what you are suggesting?). In the objective you mention it as a treatment for keloid scars and I am not sure how this will work, is it to prevent them forming?

Reply: Patients with a documented history of keloid, after injury, stem cells can be applied to the wound to prevent the formation of keloids. This is my idea

  1. Overall the paper needs some work (but is fixable) on the text in all sections of the publication. I suggest that you use your reference 9 for a guide as to how to write the results section as its is currently very brief an lacking rationales and what has been done. I have put some comments below to help you edit the paper.

Reply: Done as suggestion

  1. In the abstract there are lots of abbreviations that have not been spelled out (you always do this the first place they are written).

Reply: Modifications have been made in the full text according to your suggestions

  1. In the objective section of the abstract the last sentence is not clear. Do you want to provide an experimental basis or provide a novel proposal – it is one or the other. This is said again at line 71 in the introduction.

Reply: we provide an experimental basis, I rewrite this sentence.

  1. Still in the abstract the methods section there is an ‘and’ missing from the list of methods and also immunofluorescence microscopy is missing.

Reply: Done as suggestion .

  1. Introductions do not have figures in them. If you want to show some examples of the samples you are using then this should be in the results section.

Reply: Moved this part from the method to the result.

  1. Line 61 Chemotherapeutics generally refer to drugs used to treat cancer. I think you probably want to say therapeutics.

Reply: Done as suggestion .

  1. Line 64 is missing an ‘and’

Reply: Done as suggestion .

  1. Line 62 is a little misleading as it says clinical trials have been published about effective therapies for fibrotic diseases using mesenchymal stem cells. You were talking about keloid scars where they have not been used clinically (see Bojanic 2021, Cell and Tissue Research) so be clear about what you mean. You also cite 2 studies that are done in vitro and not even in vivo in preclinical mouse models.

Reply: we add a new reference    Falanga V, Iwamoto S, Chartier M, Yufit T, Butmarc J, Kouttab N, Shrayer D, Carson P. Autologous bone marrow-derived cultured mesenchymal stem cells delivered in a fibrin spray accelerate healing in murine and human cutaneous wounds. Tissue Eng. 2007 Jun;13(6):1299-312. doi: 10.1089/ten.2006.0278. PMID: 17518741.

Averyanov A, Koroleva I, Konoplyannikov M, Revkova V, Lesnyak V, Kalsin V, Danilevskaya O, Nikitin A, Sotnikova A, Kotova S, Baklaushev V. First-in-human high-cumulative-dose stem cell therapy in idiopathic pulmonary fibrosis with rapid lung function decline. Stem Cells Transl Med. 2020 Jan;9(1):6-16. doi: 10.1002/sctm.19-0037. Epub 2019 Oct 15. PMID: 31613055; PMCID: PMC6954714.

  1. It’s unclear from your methods exactly how the RT-PCR and western blotting were performed. Were the cells co-cultured with the stem cells in the upper chamber and fibroblast in the well, then the upper chamber removed and the fibroblast in the lower chamber analysed? There are lots of this sort of information missing so go carefully through it to make sure it is written clearly enough that someone could repeat the study.

Reply: stem cells in the upper chamber,fibroblast in the well,when we test ,removed the upper chamber ,only fibroblast in the well were test.

  1. Line 82 did you also have antibiotics and a fungicide in your media?

Reply: yes, culture medium—complete DMEM [ (Dulbecco’s Modified Eagle Media (ThermoFisher Scientific, cat. no. 11965084) with 10% fetal bovine serum (FBS, ThermoFisher Scientific, cat. no. 10438034) and penicillin (100 U/mL)/streptomycin (100μg/mL) (cat. no. 15140-148, Gibco BRL, Carlsbad, CA) ]

  1. Line 92 ‘collagen fibers begin to coarse into thick bundles’ coarse means thick so maybe you are trying to say arrange as?

Reply: yes, exactly.

  1. Line 93 ‘Finally, fibroblasts from keloid and normal skin isolation were also performed using the explant growth methodology at passages 3-5.’ You need to describe this method better and what were they cultured in etc.

Reply: Done as suggestion.

  1. Line 97 what were the cells resuspended in?

Reply: the cell were resuspended in in culture medium

  1. Line 102 Analysed by flow cytometery. This needs more details, e.g. machine and what software you used.

Reply: Done as suggestion.

  1. Line 105 you need to mention that it’s the Transwells that have the pores. The way this is written it reads as if the plates have pores. Also, the figure needs removing (Transwells are standard method) and the information should be described better in the methods text. For example you need to add how many cells were plated in the well and in the Transwell upper chamber for each cell type.

Reply: Done as suggestion.

  1. Line 121. How did you analyse and define cell morphology?

Reply: we did not analyse cell morphology

  1. Line 124. You have not mentioned the cells and how many were plated etc. These details need to be added.

Both normal and keloid fibroblasts were each individually seeded into 6-well plates at a concentration of 1×104 cells per well.

  1. Line 134, I think you need to say according to the instructions in this sentence.

Reply: Done as suggestion.

  1. Line 136 0.5 ml ‘of’ solution A. Similarly, for the two lines below an ‘of’ needs adding.

Reply: Done as suggestion.

  1. Line 143 ‘confluency ‘a’ 200ul sterile’

Reply: Done as suggestion.

  1. 146 Transwell is a brand and its not written trans well

Reply: change trans well to Transwell

  1. Line 148, please explain better how you anlysed this. For example, did you use an imageJ plug in, if not what tool did you use? Are you measuring distance or area? 

Reply: we use the area to analyse. Explain the detail in manuscript

  1. Line 156 This method is not for a western blot, it looks like it is half immunofluorescence microscopy and half western blotting. Western blotting does not start with fixing cells in paraformaldehyde and permeabilising the cells like this but immunofluorescence microscopy has this in it. You have a section for western blotting and one for immunofluorescence microscopy.

Reply: First we need to apologize. Write two articles at the same time, the method of this part is wrong.

Once again, we apologize for the carelessness in our drawing process.

  1. Line 169 you need to add details of the lysis buffer.

Reply: It has been stated in the text:  lysing cells in Trizol reagent (Life Technologies, Cat No. 15596-026)

  1. Line 187 and all the sections below need some sort of lead into what you are doing and how you are doing this. Ref 9 is probably a good example for you to look at to see what I mean. At the moment we don’t know what you have done and why or what cells were used. This information must be added before you launch into the result.

Reply:

  1. Figure 2 p6 you will need to provide higher resolution images these are fuzzy, and I cannot read the axis. Your figure legend also needs more detail about the experiment performed, number of samples analysed etc. Have a look at other papers and see how they present similar data.

Reply: We did more complete immunophenotyping for this part again. According to the minimum criteria set by The International Society for the Cellular Therapy in 2006 for defining multipotent MSCs. And submit new clear figure again.

  1. Line 199 again you need to say what you have done and maybe why you have co-cultured the cells so the person reading this knows what you are trying to do with this experiment. Same throughout the results.

Reply: we use the new title  HDPSCs did not inhibit the proliferation of HNFs and HKFs

Figure 4 the key to the graph needs better labelling – co-culturing of what? You have only one cell type in the key. Please take a look at reference 9 and see how they have put together their figures. Again all your figure legends need work.

Reply:

  1. Figure 4, the morphology part of the figure, I am not sure what I am supposed to be looking at and at the magnification provide it is hard to determine morphology. There is also no main text associated with this result so you have not told me why you have done this or what you found in your text section.

Reply: we submit big magnification figure. Cell morphology is necessary to display, you can see the approximate cell density and state

  1. The LIVE/DEAD® staining kit (Cat# L3224; Molecular Probes) you used does not measure apoptosis, only live or dead cells. You need to fix this in the text and in your figure as its not apoptosis being measured. I also cannot see any dead cells in those figures as the magnification is not high enough?

Reply: Live/dead assay can measure apoptosis, we used Live/Dead assay kit (Invi-trogen). And many other paper also use this method. Prasad S, Yadav VR, Ravindran J, Aggarwal BB. ROS and CHOP are critical for dibenzylideneacetone to sensitize tumor cells to TRAIL through induction of death receptors and downregulation of cell survival proteins. Cancer Res. 2011 Jan 15;71(2):538-49. doi: 10.1158/0008-5472.CAN-10-3121. Epub 2010 Dec 2. Retraction in: Cancer Res. 2018 Sep 1;78(17):5185. PMID: 21127198; PMCID: PMC3022089. 

Jiang Z, Li H, Qiao J, Yang Y, Wang Y, Liu W, Han B. Potential Analysis and Preparation of Chitosan Oligosaccharides as Oral Nutritional Supplements of Cancer Adjuvant Therapy. Int J Mol Sci. 2019 Feb 20;20(4):920. doi: 10.3390/ijms20040920. PMID: 30791594; PMCID: PMC6412339.

  1. Please provide quantification of all of the western blots.

Reply:

  1. The discussion mentions ‘the results of the present study revealed that dental pulp stem cells are also able to inhibit the migration of keloid fibroblasts in vitro, thereby suggesting that dental pulp-derived stem cells can promote wound healing by inhibiting scar formation.’ Please add some more discussion around why this will inhibit scar formation.

Reply: Done as suggestion.

  1. There are a few typographical errors that need fixing and some small issues with grammar.

Reply: Done as suggestion.

  1. The discussion mentions ‘this study revealed the molecular mechanism by which dental pulp stem cells inhibit the proliferation and migration of keloid fibroblasts, and whether or not dental pulp stem cells play a regulatory role by secreting cytokines’. This is not what you have shown. Both the PCR and western blot are on the fibroblasts, not the stem cells, so you have not shown they secrete cytokines. You need to look carefully at your results and your interpretation of them.

Reply: we rewrite that part

Reviewer 3 Report

The study of Ming Yan and coworkers aims to investigate the role of human dental pulp stem cells in modulating the phenotype of keloid fibroblasts.

Finding an effective treatment for keloids is relevant and of interest.

Data from the present study are of potential interest, but there are several and serious technical drawbacks

Authors should consider the following points to better focus on results and to help readers:

  • Abstract: too many acronyms without an explanation the first time they are mentioned. The abstract is supposed to be self-explanatory before reading the text
  • Line 27: Authors emphasize the reduced expression of TGF-beta 1 and 2, but they do not refer to changes in CTGF in contrast to comments in the text
  • Line 28: Authors state a decreased expression of fibronectin, however no data are reported
  • Figure 1: Place labels appropriately
  • Line 73: The manuscript does not actually provide the mechanisms of the influence of HDPSC on keloid fibroblast, as mentioned in the introduction.
  • Line 133: it has to be specified if determinations were performed after 48h from seeding or after cells reached confluence. Usually, a reliable measure is done after cells reach confluence
  • Lines 156-164: Authors have to check the WB procedure. It is not consistent
  • Line 170: Please check procedure. Lysing spirochetes??
  • Line 177: Authors measured the expression of fibrinogen. Explain the meaning of this measure or check if appropriate. Authors should refer to measures whose results are reported in the text. TGF-beta2?? Collagen??
  • Figure 4A: check labels
  • Figure 4A: Authors should explain why changes can be observed only at the 3-day time point. Please report significance values on graph
  • Figure 4B: the morphology of the cells is barely visible. Change magnification
  • Line 212: How was measured apoptosis? By live/dead staining kit? If it is the case, this method is not appropriate. Other parameters should be used  to demonstrate the occurrence of apoptosis as Annexin V, DNA degradation, caspases, Bcl2….
  • Line 219: casein is not correct
  • Line 231: Cellular migration. Why it was measured only after 24 h? Please explain why the effects on cell migration were no validated at different time points
  • Line 242: Check correspondence to materials and methods
  • Line 244: Please explain why values were normalized against tubulin.
  • Figure 8: check labels in panel B
  • Lines 247-250: Check correspondence of data in figure 8 with text
  • Line 262: Authors refer to the wrong figure
  • Line 361: The sentence that “this study revealed the molecular mechanisms… “ is definitely an over statement. Results do not actually provide any explanatory mechanisms.
  • The role of myofibroblasts in fibrotic process has been recently highly emphasized. Authors should consider the recent literature on this subject and comment more on changes in alpha-SMA. It should be of interest to investigate on the expression of myofibroblast’s markers.

Author Response

  • Abstract: too many acronyms without an explanation the first time they are mentioned. The abstract is supposed to be self-explanatory before reading the text
  • Reply: Done as suggestion
  • Line 27: Authors emphasize the reduced expression of TGF-beta 1 and 2, but they do not refer to changes in CTGF in contrast to comments in the text
  • Reply: We modified the previous confusing expression in abstract and text.
  • Line 28: Authors state a decreased expression of fibronectin, however no data are reported
  • Reply: We rewrite this part.
  • Figure 1: Place labels appropriately
  • Reply: We reproduced the figure.
  • Line 73: The manuscript does not actually provide the mechanisms of the influence of HDPSC on keloid fibroblast, as mentioned in the introduction.
  • Line 133: it has to be specified if determinations were performed after 48h from seeding or after cells reached confluence. Usually, a reliable measure is done after cells reach confluence
  • Reply: We also did it after cell confluence, so add this sentence
  • Lines 156-164: Authors have to check the WB procedure. It is not consistent
  • Reply:we rewrite this part
  • Line 170: Please check procedure. Lysing spirochetes??
  • Reply: change “Lysing spirochetes” to “lysing cells”
  • Line 177: Authors measured the expression of fibrinogen. Explain the meaning of this measure or check if appropriate. Authors should refer to measures whose results are reported in the text. TGF-beta2?? Collagen??
  • Reply: we change labels formation
  • Figure 4A: check labels
  • Reply: we change labels formation
  • Figure 4A: Authors should explain why changes can be observed only at the 3-day time point. Please report significance values on graph
  • Reply: The black and gray columns represent the proliferation of HKF; The red and pink columns represent the proliferation of HNF. Compared to the HKFs cultured alone, the CCK‐8 assay showed that no difference in the proliferation of HKFs was observed in the DPSCs/HKFs co-culture group at days 1,3,5, and 7 (P>0.05). Moreover, as a positive control, there was no detected difference in the proliferation of HNFs in the DPSCs/HNFs co-culture group at days 1,3,5, and 7 (P>0.05). However, the proliferation rate of HKFs was much higher than that of HNFs on the 3rd day (P<0.05) (Fig. 4).
  • Figure 4B: the morphology of the cells is barely visible. Change magnification
  • Reply:Done as suggestion
  • Line 212: How was measured apoptosis? By live/dead staining kit? If it is the case, this method is not appropriate. Other parameters should be used to demonstrate the occurrence of apoptosis as Annexin V, DNA degradation, caspases, Bcl2….
  • Reply: I added this answer in the method section how do we calculate: Five images, one from the center and four from the periphery, were obtained from each well using an Olympus inverted fluorescence microscope. Live and dead cells were counted using ImageJ software, and the rates of cell survival were calculated as Live cells/(Live + Dead cells)%
  • Line 219: casein is not correct
  • Reply: change Calsein-AM to Calcein-AM
  • Line 231: Cellular migration. Why it was measured only after 24 h? Please explain why the effects on cell migration were no validated at different time points
  • Reply: The concrete time of cell migration depends on cell type and cell inoculation density. In principle for migration is ensure that the density of post-migratory cells, including the control group and the experimental group, is suitable and easy to count.
  • And Numerous studies using other mesenchymal stem cells to affect keloid fibroblasts. (PMID: 27211019 、PMID:33456506 )
  • Line 242: Check correspondence to materials and methods
  • Line 244: Please explain why values were normalized against tubulin.
  • Reply: Rigorous Western Blot experimental design requires a good reference system, which is very useful for the analysis of experimental results. Tubulin is divided into α, β, γ, δ, ε and many other types of tubulin, among which α-Tubulin and β-Tubulin can form heterodimers, which are the two main types of Tubulin that form microtubules.
  • The molecular weights of α-tubulin and β-tubulin are 55kDa and 50kDa, respectively, and the actual detection bands are both around 55kDa. As an internal control, the protein level of β-tubulin usually does not change, so it is widely used as a reference for the consistency of sample loading during Western Blotting.
  • Figure 8: check labels in panel B
  • Reply: we add that label
  • Lines 247-250: Check correspondence of data in figure 8 with text
  • Reply: we rewrite that sentence
  • Line 262: Authors refer to the wrong figure
  • Reply: We corrected this error
  • Line 361: The sentence that “this study revealed the molecular mechanisms… “ is definitely an over statement. Results do not actually provide any explanatory mechanisms.

Reply:we rewrite this part

  • The role of myofibroblasts in fibrotic process has been recently highly emphasized. Authors should consider the recent literature on this subject and comment more on changes in alpha-SMA. It should be of interest to investigate on the expression of myofibroblast’s markers.

Reply:yes exactly ,you are the real expert in this filed ,if we Pass the first round ,we would like to do some experiment about myofibroblasts

Round 2

Reviewer 1 Report

Most of the comments have been taken into account. I do not fully agree with the number of repetitions, but the SD looks decent despite small research groups.

Author Response

We sincerely thank you for your review, support and constructive suggestions for improvement of our work. After carefully reading your comments, we revised our manuscript accordingly. As for the number of replicate samples, we hope to solve this fascinating problem in the future. When we collect more samples and increase animal experiments, we will submit another article.

Reviewer 2 Report

There is still no mechanism here. The authors need to show that the stem cells are secreting some of the cytokines that promote repair. You have shown results from the fibroblasts but not the stem cells. Also, how the stem cells are altering the fibroblasts should be determined.

Methods section still need a bit of work and should be written so that your experiments can be repeated. For example, for antibodies e.g. for Flow cytometry or western blotting, you need the details of where the antibody was purchased and its catalogue number if that company has more than 1 antibody to that marker.

Figure 3 should be in the main results if you are not able to describe what you have done in the results or it should be removed. The text should clearly describe the experiments.  

The results section still needs lead in sentences in places, if the reader has to go back to things like the methods or looking at figures to try to get a feel as to why you are doing in an experiment then it makes the paper harder to read. Information must be added before you launch into the result. An example might be – ‘Stem cells were collected from dental pulp using the limiting dilution method. To confirm these HDPSCs were of mesenchymal origin they were analysed for……

The LIVE/DEAD stain does not measure apoptosis. The detection of dead cells works by the stain being able to enter dead cells that have a permeabilised membrane (i.e. necrotic cells) to look at dead cells (see the manufactures web site for details on this). This is not an apoptotic cell stain as apoptotic cells have intact membranes so this dye cannot enter them. You need to stain cells differently to detect apoptosis, e.g. a Tunnel assays or Annexin V staining work well.

Figure 6 the legend says cells counted but the graph has % area. These are two different things. Please clear this up in the publication.

The information on the samples should be collated in a table that needs to be put in the publication not just in the rebuttal letter.

All of your figure legends need to have in them the exact number of samples used in each experiment, whether the error bars are SEM or SD and the statistical test that has been applied.

This previous comment has not been fixed in the publication. ‘It’s unclear from your methods exactly how the RT-PCR and western blotting were performed. Were the cells co-cultured with the stem cells in the upper chamber and fibroblast in the well, then the upper chamber removed and the fibroblast in the lower chamber analysed? This needs to be in the publication, either in the legend or the methods.

In response to ‘How did you analyse and define cell morphology?’ you say you did not analyse cell morphology. Figure 4 legend says ‘the morphology of fibroblasts was determine’ – again how did you determine morphology, what are you looking for? Round cells? Elongated? Or do you mean something other than morphology? This needs correcting and the reasons for doing this stated clear in the results text.

Figure 2 you will need to provide higher resolution images these are still fuzzy, and I cannot read the axis well. Same for Figure 4, the images need to be better. The bottom panels look out of focus and it is hard to see any cell morphology. Also you say in the results section that the the proliferation rate of HKFs was much higher than that of HNFs on the 3rd day (Fig. 4). The IF images do not really reflect this result as the cell numbers look similar. Please provide a more representative image.

Author Response

Thank you very much for your helpful review comments. We totally understood and value your comments and made a significant effort to address them. In response, the paper was significantly modified, and several new measurements were included. It is with great admiration and respect that we appreciate your meticulous revision work.

There is still no mechanism here. The authors need to show that the stem cells are secreting some of the cytokines that promote repair. You have shown results from the fibroblasts but not the stem cells. Also, how the stem cells are altering the fibroblasts should be determined.

Reply:About the cytokines, we hope to address this fascinating question in the future. We will submit another article when we collect more samples and increasing animal experiments.

Methods section still need a bit of work and should be written so that your experiments can be repeated. For example, for antibodies e.g. for Flow cytometry or western blotting, you need the details of where the antibody was purchased and its catalog number if that company has more than 1 antibody to that marker.

Reply:done as suggested.

Figure 3 should be in the main results if you are not able to describe what you have done in the results or it should be removed. The text should clearly describe the experiments. 

Reply:done as suggestion

The results section still needs lead in sentences in places, if the reader has to go back to things like the methods or looking at figures to try to get a feel as to why you are doing in an experiment then it makes the paper harder to read. Information must be added before you launch into the result. An example might be – ‘Stem cells were collected from dental pulp using the limiting dilution method. To confirm these HDPSCs were of mesenchymal origin they were analysed for……

Reply:

The LIVE/DEAD stain does not measure apoptosis. The detection of dead cells works by the stain being able to enter dead cells that have a permeabilised membrane (i.e. necrotic cells) to look at dead cells (see the manufactures web site for details on this). This is not an apoptotic cell stain as apoptotic cells have intact membranes so this dye cannot enter them. You need to stain cells differently to detect apoptosis, e.g. a Tunnel assays or Annexin V staining work well.

Reply:About the apoptosis, Indeed, you are right. we seriously concerned the fact that there is no difference in apotosis because we have erroneously used the wrong method. What should be the reasonable naming for this part of the experimental results?

we hope to address apoptosis question in the future. We will submit another article when we increase animal experiments.

Figure 6 the legend says cells counted but the graph has % area. These are two different things. Please clear this up in the publication.

Reply:Percentage of wound area recovery by migrated cells was quantified by Image J.

The information on the samples should be collated in a table that needs to be put in the publication not just in the rebuttal letter.

Reply:done as suggested.

All of your figure legends need to have in them the exact number of samples used in each experiment, whether the error bars are SEM or SD and the statistical test that has been applied.

Reply:done as suggested.

This previous comment has not been fixed in the publication. ‘It’s unclear from your methods exactly how the RT-PCR and western blotting were performed. Were the cells co-cultured with the stem cells in the upper chamber and fibroblast in the well, then the upper chamber removed and the fibroblast in the lower chamber analysed? This needs to be in the publication, either in the legend or the methods.

Reply:done as suggested. Normal and keloid fibroblasts were seeded in six-well plates at a density of 5 × 105 cells/well and transwell chambers were seeded with a concentration of 5 × 105 of dental pulp stem cells cultured in complete medium for 24 hours. The medium was discarded and the fibroblast in the lower chamber washed 3 times with phosphate buffered saline (PBS).

In response to ‘How did you analyse and define cell morphology?’ you say you did not analyse cell morphology. Figure 4 legend says ‘the morphology of fibroblasts was determine’ – again how did you determine morphology, what are you looking for? Round cells? Elongated? Or do you mean something other than morphology? This needs correcting and the reasons for doing this stated clear in the results text.

Reply:done as suggested. I deleted ‘the morphology of fibroblasts was determine’  and rewrite the result text : Compared with group control, the morphology of fibroblast is still elongated, and there is no variation between two groups.

Figure 2 you will need to provide higher resolution images these are still fuzzy, and I cannot read the axis well. Same for Figure 4, the images need to be better. The bottom panels look out of focus and it is hard to see any cell morphology. Also you say in the results section that the the proliferation rate of HKFs was much higher than that of HNFs on the 3rd day (Fig. 4). The IF images do not really reflect this result as the cell numbers look similar. Please provide a more representative image.

Reply:done as suggested.

Reviewer 3 Report

The manuscript has been revised taking into consideration  most of reviewers' suggestions and comments. Still few minor points need to be checked:

1) Line 28-29: check spelling and correct acronym

2) line 34  and 415: Authors state "promoting the expression of anti-fibrotic genes". No mention in the results of anti-fibrotic gene measurements.

Author Response

Thank you very much for your helpful review comments. We totally understood and value your comments and made a significant effort to address them. It is with great admiration and respect that we appreciate your meticulous revision work. In response, the paper was significantly modified, and two more new questions were included.{1) Line 28-29:  2) line 34  and 415: }. 
